# Detection of *Echinococcus multilocularis* in coyotes in Washington State, USA highlights need for increased wildlife surveillance

Yasmine Hentati [1,2]*, Ellie Reese[2], Claire C. Curran[3], Erika M. Miller[4],
Dakeishla M. Díaz-Morales[5], Samantha E.S. Kreling[2,6], Guilherme G. Verocai[7],
Laura R. Prugh[2], Christopher J. Schell[1], Chelsea L. Wood[8]

1 Department of Environmental Science, Policy, and Management, University of California, Berkeley, California, United States of America, 2 School of Environment and Forest Sciences,University of Washington, Seattle, Washington, United States of America, 3 Batten School of Coastal and Marine Sciences, College of William & Mary, Williamsburg, Virginia, United States of America, 4 Sound Data Management LLC, Seattle, Washington, United States of America, 5 Department of Biological Sciences, DePaul University, Chicago, Illinois, United States of America, 6 Department of Forestry and Natural Resources, Purdue University, West Lafayette, Indiana, United States of America, 7 College of Veterinary Medicine & Biomedical Sciences, Texas A&M University, College Station, Texas, United States of America, 8 School of Aquatic and Fishery Sciences, University of Washington, Seattle, Washington, United States of America

* yhentati26@gmail.com

## Abstract

*Echinococcus multilocularis* is a zoonotic cestode that uses canids as definitive hosts and rodents as intermediate hosts. In humans, this parasite is the causative agent of alveolar echinococcosis. Recently, its range has been expanding across the Northern Hemisphere, and it is increasingly detected in wild canids, domestic dogs, and humans across Canada and the United States. While this expansion has been documented in isolated studies across the continent, a lack of routine sampling in wildlife hinders our ability to anticipate and mitigate further spread of *E. multilocularis*. We confirmed the presence of *E. multilocularis* in Washington State, USA, using a combination of morphological and molecular techniques across carcasses and field-collected scats of coyotes (*Canis latrans*), this region's most common wild canid. Morphological identification of adult worms was confirmed by next-generation sequencing. Over a third of all samples tested positive for *E. multilocularis* when all methodologies were combined. Sequencing revealed a haplotype of *E. multilocularis* matching a documented haplotype originally of European origin in British Columbia, Canada. Our study provides the first confirmation of *E. multilocularis* in a wild host on the west coast of the contiguous United States and provides additional haplotype information crucial to tracking the geographical expansion of the parasite. We also provide a new next-generation sequencing primer targeting cestodes of canids. The difference in amplification between intestinal and fecal samples suggests that non-invasive fecal sampling using DNA metabarcoding - a popular method of helminth surveillance - may lead to underestimation of prevalence, hindering control

**Data availability statement:** Data and code produced for this manuscript is available at https://github.com/yasminehentati/emulti_2025. Locations, gross necropsy data, and parasitological data from coyotes in this study are available via the University of Washington Burke Museum of Natural History Mammalogy Collection under the preparator code YH (https://www.gbif.org/dataset/830eb5d0-f762-11e1-a439-00145eb45e9a).

**Funding:** This research was funded by the US National Science Foundation grant from the Division of Environmental Biology under grant number 2223973 to LRP, CJS, and CLW and by a University of Washington Hall Conservation Genetics Fund grant to YH. YH was supported by the National Science Foundation Graduate Research Fellowship under DGE-214-0004. CLW was funded by a CAREER Award from the NSF under DEB-2141898. The funders had no role in study design, data collection and analysis, decision to publish, or preparation of the manuscript.

**Competing interests:** The authors have declared that no competing interests exist.

measures. The global significance of these findings extends beyond North America; *E. multilocularis* is a major public health concern in Europe and Asia, where alveolar echinococcosis is increasingly diagnosed in humans. Our study highlights the urgent need for increased surveillance and improved diagnostic strategies worldwide, particularly in regions with significant human-wildlife contact.

## Author summary

Parasites that are transmitted among wildlife, domestic animals, and people are an important part of global health. One such parasite is *Echinococcus multilocularis*, a small tapeworm of canids that can cause a severe, life-threatening disease in humans called alveolar echinococcosis. Many wild canid hosts of the parasite, such as coyotes, overlap significantly with domestic dogs, which facilitates transmission to humans. In Europe, Asia, and Arctic regions of North America, *E. multilocularis* has long been recognized as a major public health problem. In recent decades its range has expanded across the Northern Hemisphere, raising concern. In this study, we discovered *E. multilocularis* in coyotes in a densely populated area of Washington State, USA - the first detection of *E. multilocularis* in a wild host in the region. More than one-third of our coyote samples contained *E. multilocularis*, confirming that it is widespread in the area. Genetic testing showed that the strain we detected matched one previously found in Canada, originally from Europe. Our findings underscore the importance of monitoring *E. multilocularis* and other parasites in wildlife so that emerging public health threats can be detected early, reducing risk to people and pets.

## Introduction

The parasite *Echinococcus multilocularis* - a cestode of canids that can cause lethal disease in humans - has expanded its range in the past several decades [1,2]. It is increasingly found in temperate regions, and the parasite's expansion is thought to be driven by multiple factors, including human-facilitated animal movement [3,4], climate change [5], and urbanization [6], which have facilitated establishment in new environments and host species across the Northern Hemisphere [7–9]. As a result, *E. multilocularis* has become a growing concern not only in traditionally endemic areas, but also in regions and countries where it was once considered rare or absent (Figs 1 and S1 [10–18,19–21];). Given the geographically expansive distribution of canid species - the definitive hosts of *E. multilocularis* - and relatively high prevalence of the parasite in canids across the Northern Hemisphere [22,23], zoonotic spillover to humans and their pets occasionally occurs, though the true distribution of human infections is likely underestimated due to a lack of dedicated surveillance [22]. Unlike some of its close relatives (e.g., *E. granulosus*), the life cycle of *E. multilocularis* is primarily sylvatic (i.e., transmitted primarily among wild animal hosts

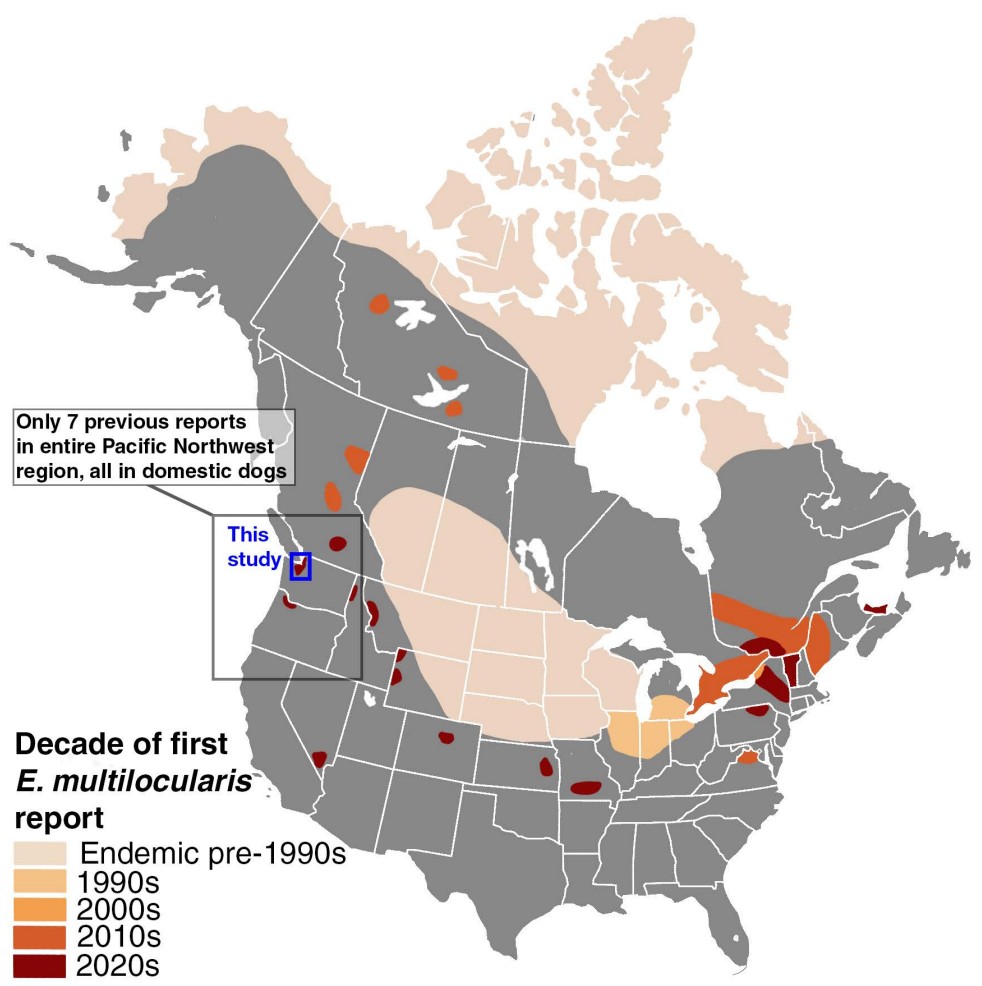

**Only 7 previous reports in entire Pacific Northwest region, all in domestic dogs**

**This study**

**Decade of first *E. multilocularis* report**

- Endemic pre-1990s
- 1990s
- 2000s
- 2010s
- 2020s

**Fig 1. A map of the United States and Canada depicting the expansion of E. multilocularis over the last several decades.** Each color represents a different decade wherein a peer-reviewed study first confirmed the presence of E. multilocularis morphologically, molecularly, or both. In cases where only one or a few isolated reports exist (e.g., the western U.S.), polygons have been expanded slightly beyond the actual range of detection for visibility. Methods for creating this map can be found in S1 Text. A similar map of Europe and Asia can be found in S1 Fig. The base map for this figure is available via Wikipedia Commons at the following link: https://commons.wikimedia.org/wiki/File:North_America_blank_map_with_state_and_province_boundaries.png.

[23,24,25];). Definitive host species include coyotes (*Canis latrans* [26];), gray wolves (*Canis lupus* [13];), red foxes (*Vulpes vulpes* [15];), Arctic foxes (*Vulpes lagopus* [26];), golden jackals (*Canis aureus* [17];), and domestic dogs (*Canis lupus familiaris* [27];). In the rodent intermediate host, *E. multilocularis* encysts in the liver and produces alveolar hydatid cysts (Fig 2), which present as cancer-like mass lesions on the surface of the liver and cause severe pathology. Wild rodents are the primary intermediate hosts for this parasite [19]. While domestic dogs and wild canids are generally definitive hosts, they can occasionally also act as aberrant intermediate hosts [28,29]. Humans can also act as aberrant intermediate hosts for *E. multilocularis* and other species within the genus *Echinococcus* (Fig 2 [21];).

The diseases caused by *Echinococcus* spp. are collectively known as echinococcosis. Echinococcosis is classified by the World Health Organization (WHO) as one of 20 Neglected Tropical Diseases, a diverse set of conditions that affect millions of people worldwide, especially from impoverished and marginalized communities, and are marked by low

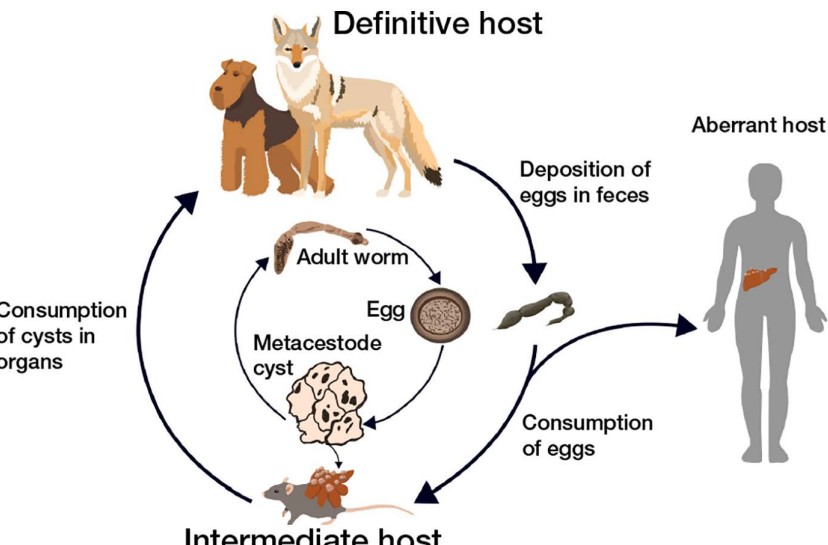

**Fig 2. Typical life cycle of *Echinococcus multilocularis*, with definitive host (canids), intermediate host (rodent), and an aberrant intermediate host (human) pictured.**

mortality rates but high morbidity. Alveolar echinococcosis (AE), caused by *E. multilocularis* infection, is considered to be the world's third most significant foodborne parasitic disease based on multicriteria joint ranking by the Food and Agriculture Organization of the United Nations and the WHO [30]. People are infected via the accidental consumption of eggs from the environment or contaminated vegetables, fruits, or water [31]. Most human cases of AE originate in rural regions of Russia and China [22,21]. Globally, there are at least 18,000 new cases of AE per year, with an estimated annual burden of 688,000 disability-adjusted life years [31].

In North America, AE was extremely rare prior to the 2010s. Most diagnosed cases were among indigenous communities on remote islands of northwestern Alaska, attributed to the North American tundra haplotype of *E. multilocularis* [11]. However, a rapid expansion of *E. multilocularis* in wild canid hosts across North America has occurred throughout the past decade, most notably in the Canadian province of Alberta, though the first detection was in a dog in central British Columbia (Fig 1 [10–18];). Many of these new populations share lineages with European populations, with definitive host infections recorded primarily in coyotes and domestic dogs at more southern latitudes [2,10,32,33]. There is considerable debate as to how these haplotypes arrived in North America and whether they should be considered "European" at all, as they are now the most common endemic haplotype on the continent [33,34]. Nonetheless, "European-type" and "Asian-type" haplotypes are thought to be more virulent and pathogenic than those native to the tundra and north-central regions of North America, which may explain the rapid establishment in sylvatic systems and emergence in humans in a relatively short period of time [11,35]. Indeed, there has been a marked increase of human cases of AE in North America, particularly in the Canadian province of Alberta, which has reported at least 17 confirmed human cases since 2013, and where *E. multilocularis* is prevalent in coyotes [11,36,37].

While the north-central historic range of *E. multilocularis* covers parts of Alberta, Saskatchewan, and Manitoba, as well as the majority of the Midwestern states, recent reports of *E. multilocularis* span across both countries (Fig 1). Canadian provinces and U.S. states that have reported *E. multilocularis* attributed to "European" types in either human or non-human hosts include Alberta [2], British Columbia [32], Ontario [38], Quebec [38], New York [12], Vermont [35], and Virginia [39]. *E. multilocularis* has increasingly been reported in coyotes, red foxes, dogs, and humans in eastern and central North America, but only a few isolated reports in dogs exist west of the Rocky Mountain Range. While diagnoses

of domestic dogs are often the first indication that *E. multilocularis* is present in a region, dogs are rarely the primary reservoir [19]. It is therefore extremely difficult to determine risk factors to dogs or humans or mitigate the spread of *E. multilocularis* without knowing the primary definitive wild host species and the prevalence rate of the parasite. However, to the best of our knowledge, no study has surveyed *E. multilocularis* in wildlife on the west coast of the United States (Washington, Oregon, and California).

Recent reports of *E. multilocularis* in dogs in the northwestern United States suggest that its distribution is much larger than previously known, but until now, these isolated detections were not linked to an established sylvatic transmission cycle. Since 2023, there have been 5 reports of *E. multilocularis* in domestic dogs in Washington state – 1 dog with AE reported in eastern Washington [40], and 4 definitive host dogs reported in western Washington [33] - in addition to 1 report each in the neighboring states Idaho and Oregon and in south-central British Columbia [33]. Despite these 7 diagnoses spanning an area of over 550,000 square kilometers (Fig 2), and the occurrence of three suitable wild canid host species in this region, no study has confirmed its presence in wildlife. Here, we present the first report of this species in a wild host on the west coast of the United States, confirming the presence of *E. multilocularis* in free-ranging coyotes in Washington State. We salvaged coyote carcasses and field-collected scat samples to screen *E. multilocularis* using both morphological and molecular methods. We developed a new next-generation sequencing cestode primer tested on various canid cestode species and used it to identify *E. multilocularis* in our samples. Additionally, we compared sequences of three key target genes to determine the haplotype of *E. multilocularis* worms in our study area and relatedness to previously published haplotype sequences.

## Methods

### Ethics statement

Carcasses were salvaged from state and federal lethal management operations, city and state agency roadkill reports, wildlife rehabilitation centers, and opportunistic roadkill collection under a Washington Department Fish & Wildlife Scientific Collection Permit (#22-263) between 2021 and 2024.

### Study area and host species

The study area lies within the Puget Sound Lowlands region of western Washington, USA (hereafter Puget Sound region), a glacially formed trough characterized by a temperate marine climate (Fig 3) [41]. The region is one of the most densely populated regions of the Pacific Northwest, with the city of Seattle (population approximately 800,000) serving as the epicenter of the metropolitan region [42]. The total population of the Puget Sound region, encompassing King, Pierce, Snohomish, Kitsap, Island, Skagit, and counties, is approximately 4.7 million [43]. Three potential wild hosts exist for *E. multilocularis* in Washington state: coyotes, red foxes, and gray wolves. Coyotes are present throughout the state and are widespread [44]. Cascade red foxes (*Vulpes vulpes canadensis*) are found in western Washington, but they are much rarer than coyotes, and were listed as endangered in 2022 [45]. Red foxes are also located on San Juan Island in the Salish sea, where they were introduced in the 20th century to control European rabbit populations [46]. Gray wolves naturally repopulated Washington in the late 2000s after extirpation in the early 20th century, but their range is generally limited to north-central and northeast Washington; however, the ranges of coyotes and gray wolves heavily overlap in this region [47,48]. Our study focused on coyotes, as they are most common in our study area.

### Coyote carcass collection

Carcasses were stored in -20°C freezers until necropsy. Gross necropsies were performed at the University of Washington Burke Museum of Natural History and Culture biology specimen preparation laboratory, a Biosafety Level 2 laboratory. Lab coats and nitrile gloves were worn during necropsies. Gastrointestinal tracts were tied off at the stomach and rectum, carefully removed with gloved hands, placed into zip-top bags, and saved for further dissection.

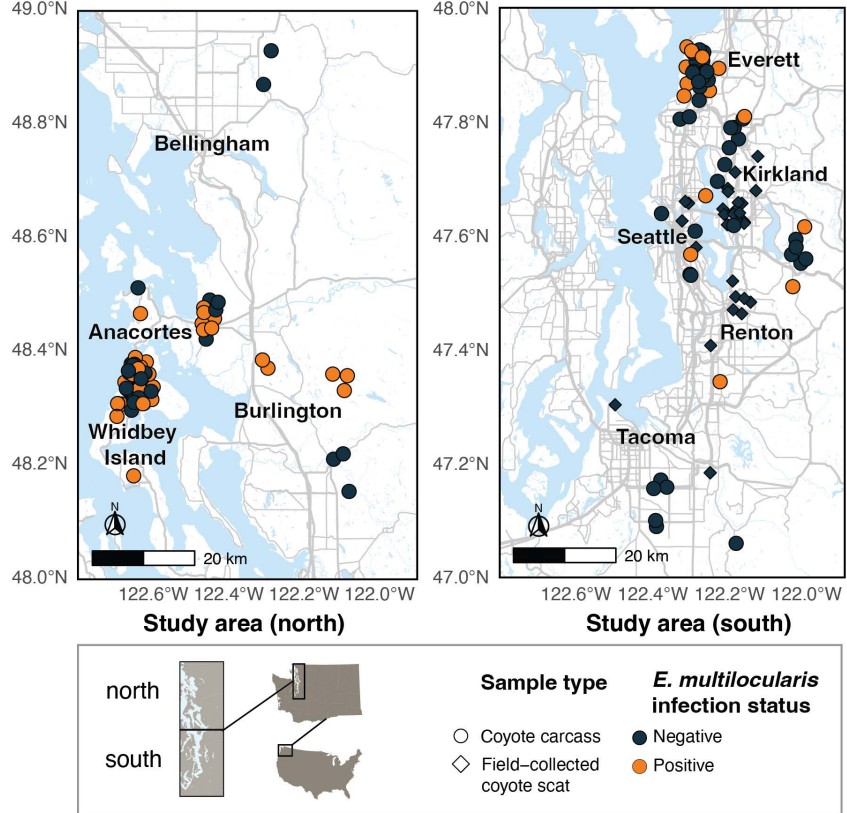

**Fig 3. Study area map showing locations of coyote carcasses and field-collected scats in this study, and the infection status of each sample. Points are jittered to better represent multiple individuals collected from the same location.** The study area is broken up into two sub-areas, north and south, for ease of map viewing. Note that no field-collected coyote scats tested positive for *E. multilocularis*, but several carcasses from the same region where scats were collected did test positive. This map was created using program R with a basemap from OpenStreetMap: https://www.openstreetmap.org/copyright.

## Coyote scat collection

Field-collected scats were obtained in Seattle and its surrounding suburbs as part of a concurrent study between 2021 and 2023 [49]. Ten samples from scats of 3 different estimated freshness categories (estimated <24 hours old, estimated 24–48 hours old, and estimated 48+ hours old) were selected. Scats were stored at -20°C after collection, with a minimum period of one week in a -80°C freezer to inactivate *Echinococcus* eggs [50]. Scats were then broken apart into 4 pieces and swabbed externally and internally with a flocked swab dipped in phosphate-buffered saline for dietary DNA metabarcoding and coyote individual genotyping. DNA was extracted following a modified version of the QIAmp DNA Investigator Kit protocol (Qiagen, Hilden, Germany). To identify individuals, scats were individually genotyped using 12 microsatellite markers (for genotyping methods, see [49]).

## Carcass dissection and worm collection

Post-necropsy, whole gastrointestinal (GI) tracts were frozen at -80°C for at least one week to inactivate parasite eggs [51]. After thawing, GI tracts were dissected using a modified version of the scraping, filtration, and counting technique (SFCT) [52], the most sensitive method for recovering *Echinococcus* spp. with minimal intestinal debris. A video recording of our modified protocol can be found in S2 Text. First, the small and large intestines were cut into 30–45 cm sections

for ease of handling. Then, each section was cut longitudinally to expose the inside of the intestine. The intestines were scraped manually by pinching the opened intestinal section between two fingers, and contents were collected in a stainless-steel bowl. This scraping was done twice for each section. Before scraping the colon, fecal samples were collected from the rectum (or as close to the rectum as possible) and stored in a Whirl-Pak bag. All remaining intestinal contents were manually mixed in the bowl and swabbed with a flocked swab dipped in phosphate-buffered saline; the tip of the swab was then stored in a 1.5 ml tube. The intestinal tissue contents were then rinsed in water to dislodge any other debris. Next, we carefully poured the intestinal contents over two stainless steel mesh sieves – a 1 mm sieve stacked over a 150 µm sieve – over a sink. We rinsed the bowl with water over the top of the 1 mm sieve until no debris was left. The sieves were then thoroughly sprayed with water to further encourage flow-through. Sieved contents were then carefully poured into two separate clean bowls, using 70% ethanol in a squeeze bottle to dislodge any stuck debris. Contents were transferred into plastic containers. We ensured that all contents were submerged completely in 70% ethanol before sealing the containers. All dissection equipment was saturated with 10% bleach for several minutes and then washed with dish soap before handling the next sample. Gloves were also changed between samples. Intestinal swabs and fecal samples were returned to -20°C until further molecular analyses. Helminth parasites were collected and stored in 70% ethanol at room temperature until morphological identification.

## Worm morphological identification

Morphological identification of adult parasites was performed using a stereoscope and a compound microscope. A subset of adult *E. multilocularis* specimens (whole worms and representative gravid proglottids) were stained with Semichon's acetic carmine, then de-stained with hydrochloric acid. Specimens were then dehydrated in a series of ethanol rinses (70%, 80%, 95%, and 100%), cleared in xylene, and mounted in Canada balsam [53]. Photos were taken using a Leica DM 2500 LED microscope and Leica microscope imaging software (LAC 4.9). Helminths were identified after published reference manuals [54] and the World Organization for Animal Health [55]. Adult cestodes of *E. multilocularis* were differentiated from those of *E. granulosus* based on total length of the adult worm, the ratio of the length of the gravid proglottids to the total length, the location of the genital pore within each proglottid, and the shape of the uterus [54,55]. We did not use egg morphology to identify *E. multilocularis* in fecal samples because its eggs are morphologically identical to those of other common canid cestodes in the family Taeniidae, including *E. granulosus* and various species of the genus *Taenia*.

## Worm and egg molecular identification

DNA was extracted from coyote samples for both DNA metabarcoding and Sanger sequencing pipelines. Cestode presence in carcass-collected fecal samples, intestinal swabs from carcasses, and field-collected scats was determined via DNA metabarcoding. After *E. multilocularis* was confirmed via DNA metabarcoding, haplotype was determined via Sanger sequencing of individual worms from intestines.

## DNA extraction (DNA metabarcoding)

Fecal samples from carcasses were first manually homogenized, then underwent 5 freeze-thaw cycles to break down protective helminth egg casings and purified using the QIAamp DNA Investigator Kit [56,57]. Intestinal swabs from carcasses were also extracted for parasite DNA using the QIAamp DNA Investigator Kit, but did not undergo further repeated freeze-thaw cycles after dissection. Presence of adult *Echinococcus* adult worms from the carcasses were compared to the PCR success rate of the primers on the scat and intestine samples (via presence/absence of DNA bands in the gels). Field-collected scat samples were previously extracted for diet metabarcoding as part of a concurrent project (see "Coyote scat collection;" [58]). To determine if our diet DNA extraction method was sufficient for capturing parasite DNA (thus saving time and costs incurred from re-extractions), a selection of these extracted samples was also prepared for sequencing.

To determine the sensitivity of swabs compared to fecal matter, we conducted an additional analysis comparing sequence read recovery of intestinal swabs, fecal matter, and fecal swabs of randomly selected carcasses (S2 Table). Fecal swabs were equally as sensitive as fecal matter for the closely related taeniid *T. pisiformis*, and slightly more sensitive than fecal matter for *E. multilocularis* (S2 Table).

## Primer development (DNA metabarcoding)

We searched the literature for cestode-specific primers that would allow for species identification via metabarcoding. However, the existing primer pairs that we found generated long amplicons (300 + bp). Longer amplicons are hard to amplify from degraded fecal DNA, lowering detection rates, and are not suitable for some sequencing platforms. We therefore developed a new cestode-specific primer pair for this study (Table 1). The resulting primer pair targets 145–147 bp of DNA from the mitochondrial 12S rRNA region of DNA of cestodes and contains species-specific single nucleotide polymorphisms (SNPs), enabling species identification via metabarcoding. We designed a forward primer to pair with a reverse primer designed by [59] (Cest5). The primers were tested on positive controls from various canine cestode species (S1 Table and S1 Data).

PCR amplification was conducted in 20 µL reactions containing 1X Qiagen Multiplex PCR Mastermix, 1X Q solution, 0.2 µM of each primer, and 2 µL of sample DNA. Thermal cycling conditions consisted of a 95°C hot-start step for 15 minutes, followed by 8 touchdown cycles of 94°C for 30 s, 59-0.5°C for 30 s, and 72°C for 30 s. This was followed by 35 cycles of 94°C for 30 s, 55°C for 30 s, and 72°C for 30 s, followed by a final extension step of 60°C for 15 min. PCR and extraction negatives were amplified alongside the samples to monitor possible cross-contamination. The resulting products were visualized on a 1% agarose gel to confirm that the primers amplified fragments of the target base pair size.

## Molecular analysis (DNA metabarcoding)

TW-F1 and TW-R1, modified to include adapters for Illumina sequencing, were used to amplify DNA. In preparation for metabarcoding, PCR amplicons were cleaned using SPRI beads, quantified using the Qubit quantification platform, normalized, and pooled. Samples were sequenced at the University of Washington Northwest Genomics Center using an Illumina NextSeq machine.

Successfully sequenced samples were analyzed using CLC Workbench software. Sequence data was demultiplexed, filtered and trimmed, and clustered by operational taxonomic units. Resulting sequences from carcass and field-collected samples were compared against the publicly available GenBank database and positive control samples. While triplicate reactions would have been ideal, this sequencing run was originally conducted as part of research and development on helminth NGS primers and we therefore chose to run only a single reaction per sample to save costs. As we only conducted one PCR replicate per sample, we used a conservative sequence read cutoff of 1,000 read counts.

## DNA extraction (Sanger sequencing)

To identify the haplotype of *E. multilocularis* present in our samples, we collected single worm samples from coyote carcasses. Samples selected for haplotype analysis were limited to 6 carcasses with both positive molecular and positive morphological IDs at the time of Sanger sequencing preparation. From these 6 samples, we extracted DNA using

**Table 1. Cestode primer pair used in this study to sequence *E. multilocularis* with next-generation sequencing. The forward primer was designed for this study. Reverse primer is from 62 (Cest 5).**

| Primer | Primer sequence (5'-3') |
| --- | --- |
| TW-F1 | TTAAGCYAAGTCTATGTGCTGC |
| TW-R1 | GCGGTGTGTACMTGAGCTAAAC |

**PLOS Neglected Tropical Diseases**

Qiagen's tissue extraction kit, then PCR amplified them in five singleplex reactions using the DNA primer pairs from Nakao et al. 2009. Extraction protocols followed those described in Nanao et al 2009. PCR conditions were modified as follows: PCR was conducted in 20 µL reactions containing 1X Q5 hotstart master mix (NEB), 1X Q solution (Qiagen), 0.2 µM of each primer, and 2 µL of sample DNA. Thermal cycling conditions consisted of a 98°C hot-start step for 3 min, followed by 7 touchdown cycles of 98°C for 10 s, 67–1°C for 60 s, and 68°C for 105 s. This was followed by 40 cycles of 98°C for 10 s, 60°C for 60 s, and 68°C for 105 s, followed by a final extension step of 72°C for 15 min. Negative extraction and PCR controls were used to monitor contamination.

### Molecular analysis (Sanger sequencing)

PCR products were analyzed via gel electrophesis to determine if amplification was successful. We included extraction and PCR negative controls, and none of our negative controls showed signs of contamination/amplification. PCR products with positive gel results were then sent to GeneWiz (Azenta) for Sanger sequencing, where they were sequenced in both the forward and reverse directions. Sanger sequencing data were quality-checked and final sequences were compiled in MEGA (MEGA11). The resulting sequences were quality checked and aligned with reference sequences from previous *E. multilocularis* studies to determine haplotype. We targeted the genes *cob* (1331 base pairs), *cox1* (1068 bp), and *nad2* (882 bp) for haplotype identification based on existing sequence data available for comparison.

To determine the haplotype of our *E. multilocularis* specimens, we aligned sequences of the three target genes on MEGA7 using publicly available sequences from GenBank and previously published studies on *E. multilocularis* (S2 Data). We used an *E. granulosus sensu stricto* sequence as an outgroup. Using the R package *pegas*, we assigned haplotypes to each sequence and built a haplotype network.

## Results

The total number of individuals represented across all sample types was 100. Of these, 37 coyotes were positive for *E. multilocularis* using either next-generation sequencing or morphological identification (18 from Whidbey Island, 8 from Everett, 4 from Burlington, 2 from Seattle, and 1 each from Sammamish, Issaquah, Auburn, Bothell, and Anacortes). In total, 63 coyotes were negative across all sample methods.

### Molecular results

We dissected GI tracts and/or morphologically identified worms from 81 coyotes across King, Snohomish, Pierce, Skagit, and Island Counties in Washington. We sequenced fecal samples and intestinal swabs from a subset of 32 individuals. Across these 32 individuals, we had a total of 32 fecal samples and 22 intestinal swabs. Ten carcasses had a fecal sample but no intestinal swab and 21 carcasses had both a fecal sample and intestinal swab. These discrepancies were due to changes in protocol partway through the study (i.e., we began collecting intestinal swabs after data collection had begun) and condition at necropsy (i.e., some carcasses were too decomposed for a full necropsy and GI tract dissection, but were sufficient for the collection of fecal samples). We also sequenced 30 coyote scat samples collected from the field. Across these field-collected scats, we identified 19 additional unique individuals using microsatellite markers. *E. multilocularis* was not detected in any field-collected scat samples, but *Taenia pisiformis*, a closely related taeniid with morphologically identical eggs to *E. multilocularis*, was successfully sequenced in 43% (n = 13) of these samples. Using only carcass samples, 81 individuals were represented, and 45.7% (n = 37) coyotes were infected with *E. multilocularis*. Across all methods, 100 total individuals were represented, and 37% of coyotes (n = 37 individuals) were infected. A summary of sample types, sample sizes, and percent of individuals infected can be found in Table 2, and a breakdown of sample sizes and infections by sample type can be found in Table 3.

**Table 2. Summary of E. multilocularis prevalence in our study, split by sample type (carcass or field-collected scat).**

|  | Carcasses | Field-collected scats | Total |
|---|---|---|---|
| Number of individuals | 81 | 19 | 100 |
| *E.m.* positive | 37 (45.7%) | 0 (0%) | 37 (37%) |

**Table 3. Table depicting each sample type, sample sizes, and number and percent of Echinococcus multilocularis infected individuals (carcasses only) in this study. Note that some individuals are represented across multiple sample types, and therefore the total number of individuals is less than the total number of samples. To calculate the number and percent of infected individuals, an individual was considered infected if at least one of their sample types was positive.**

| Sample type | Sample size | Number infected with *E. multilocularis* | Percent infected with *E. multilocularis* |
|---|---|---|---|
| Fecal sample from carcass | 32 | 14 | 43.8% |
| Intestinal swab from carcass | 22 | 17 | 77.3% |
| Morphological ID from carcass | 76 | 33 | 43.4% |
| **Unique individuals across carcass samples only** | 81 | 37 | 45.7% |

*E. multilocularis* DNA amplified more reliably from the intestine swabs than from the extracted fecal samples from the same coyotes, and intestine swabs had higher agreement with morphological data. One-third of infections were detected with intestinal swabs and/or morphological ID, but not with fecal samples. For individuals that had both intestine swabs and morphological data, these methods were equally sensitive, each detecting the same number of infections (n = 22). For individuals that had both fecal samples and morphological data, morphological identification was more sensitive—26% of infections were detected with morphological identification but not with fecal sample DNA (n = 7 of 27). Both our scat and carcass data suggests that *E. multilocularis* is more prevalent in the northern part of our study area (Fig 3). However, recent detections of *E. multilocularis* in domestic dogs from the southern Puget Sound region as well as from northern Oregon indicate that *E. multilocularis* is indeed present further south [33].

From the 6 coyotes selected for haplotype analysis using the *nad2*, *cob*, and *cox1* genes, 2 did not have successful gel results, likely due to DNA degradation during storage. PCR products from the 4 individuals that had positive gel results were sent for Sanger sequencing at Genewiz (Azenta). After aligning our concatenated sequence with others from the literature and trimming null values, the total haplotype sequence length was 3178 bp (S2 Data). Our haplotype matched a previously discovered haplotype in British Columbia (GenBank accession numbers PX737897, PX737898, PX737899; matching sequences in 29,63). According to our haplotype network, this haplotype is most closely related to *E. multilocularis* haplotypes from France (mtG h1, also known as E3).

## Morphological results

We cleared, mounted, measured, and photomicrographed representative individuals from three representative coyote hosts (Fig 5). Individuals measured between 1.2 and 3 mm (Fig 5). Gravid proglottids were less than half the total body length in all measured specimens. The genital pore locations in our worms were clearly anterior to the midpoint of the proglottid. Finally, eggs within gravid proglottids of photographed specimens outlined the sac-like shape of the uterus. The total lengths, gravid proglottid/total length ratio, genital pore location, and uterus shape of our specimens (Fig 5) are consistent with previously defined morphological characteristics of *E. multilocularis* [54,55].

## Discussion

### Summary of findings

Our findings suggest that *Echinococcus multilocularis*, a zoonotic parasite, may be far more widespread than previously recognized. When we began data collection for this study, *E. multilocularis* had not been reported in the state of

Washington or its neighboring states. Its close relative *Echinococcus granulosus* had been incidentally reported in both canine definitive hosts and ungulate intermediate hosts (K. Mansfield, Washington Department of Fish and Wildlife; personal communication). We therefore expected to find *E. granulosus* rather than *E. multilocularis* in our samples. However, since 2023, *E. multilocularis* has been detected in 5 dogs in Washington [40], including in the Puget Sound region [33]. While the lack of reports until 2023 may suggest a recent establishment of the parasite in the region, the high prevalence in our coyote population indicates that *E. multilocularis* had previously established a robust transmission cycle since at least 2021, when our sample collection began. Our haplotype sequences matched mtG h28 (in some studies referred to as BC1; Fig 4), first reported in a dog in Quesnel, British Columbia, Canada (located west of the Canadian Rockies and east of the Canadian Pacific Coast mountains) and in coyotes in east-central British Columbia [2]. To the best of our knowledge, this haplotype has not yet been reported in wildlife outside of Canada [29,64]. Our results point towards a shared common ancestor between *E. multilocularis* individuals in western Washington state and central British Columbia, despite these locations (Quesnel and Seattle) being almost 800 km and a mountain range apart.

### The promise and pitfalls of molecular surveillance

Currently, two preferred methods exist for surveying *Echinococcus* spp. in wildlife: morphological identification from adult worms from GI tracts and molecular identification from fecal samples [23,65]. Morphological identification from fecal

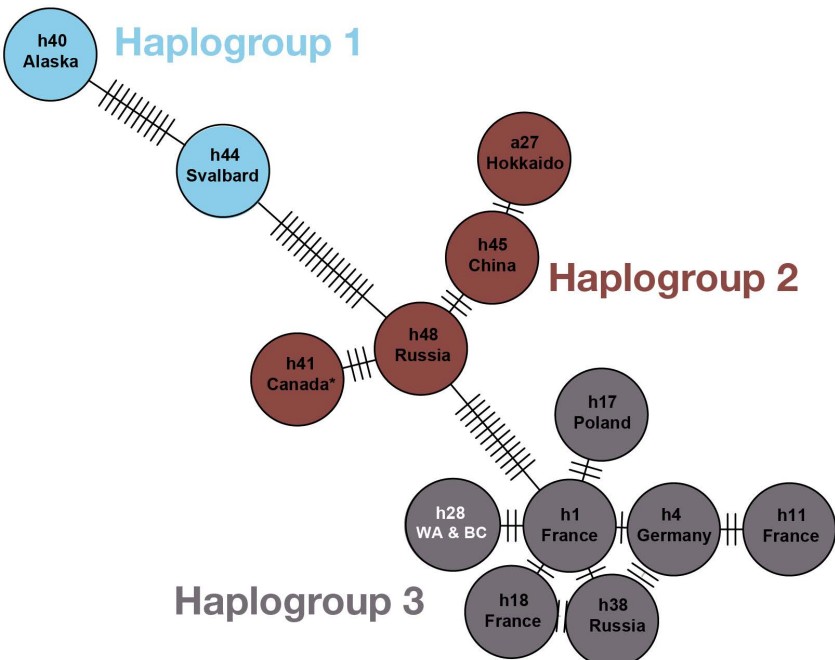

**Fig 4. Haplotype network constructed using the pegas package in R with the cob, cox1, and nad2 genes, showing some of the global published haplotypes of E. multilocularis and using the haplotype grouping system recently published in[61].** GenBank accession numbers for each haplotype from the literature are available in Table 1 of 64. Each circle represents one haplotype with names as listed in [61]; full names in Lallemand et al. include the prefix "mtG"). Country or region names below haplotype names indicate where the specimen that was sequenced came from. Tick marks between circles represent mutational steps (i.e., single nucleotide polymorphisms) that separate each haplotype from its relatives. The haplotype network is color-coded using haplotype groups from [61]. The haplotype from our study (GenBank accession numbers PX737897, PX737898, PX737899; labeled h28 WA & BC) is indicated in white and was first published in [28,62] from a dog in Quesnel, British Columbia, Canada (100% base pair match between our sequences and that from [62]. This haplotype is referred to as BC1 in some publications [2,32] and is most closely related to the mtG h1 haplotype from France (only two base pairs apart). Another haplotype from Canada (unknown region) in Haplogroup 2 is marked with an asterisk, first published in [63].

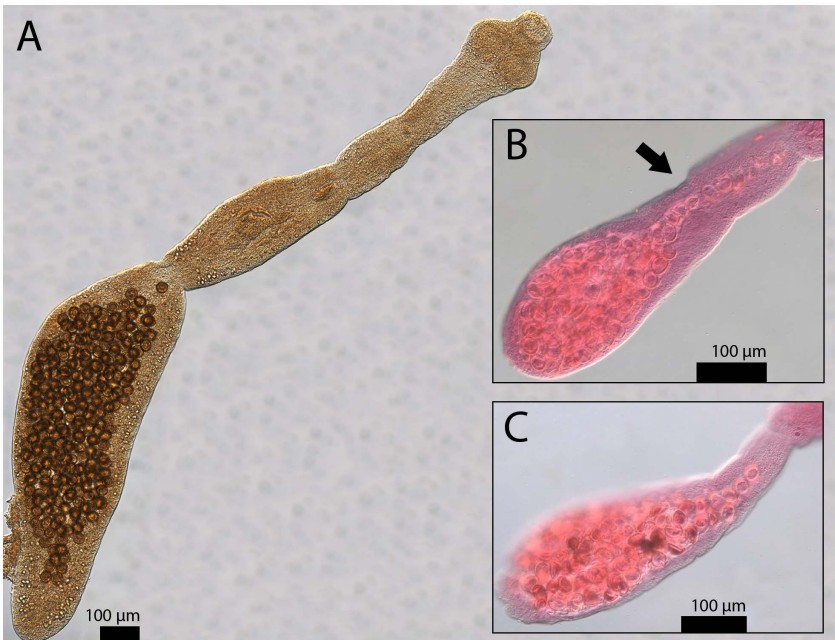

**Fig 5. Representative specimens of E. multilocularis with photos of an unstained whole worm (A), a stained gravid proglottid (B), and a second stained gravid proglottid (C).** These photos demonstrate identifying characteristics of *Echinococcus multilocularis*, especially in comparison to *Echinococcus granulosus*, a close relative ([54], see [60] for direct comparisons between the two species). Scale bars are included for each image. In A, the gravid proglottid is less than half the length of the total body length (it is more than half in E. granulosus). The total length of the specimen in A is 2.22 mm (2,221.356 µm) and its gravid proglottid is 1.07 mm (1,072.179 µm). An arrow points to the genital pore locations in **B.** The genital pore is anterior to the midpoint of the proglottid (it is posterior to the midpoint in *E. granulosus*). The sac-like uterus shape can also be seen in A, B, and C (E. granulosus has a laterally branching uterus). Photo backgrounds have been edited and arrows added to improve clarity of photos (unmanipulated versions of A, B, and C can be found in S2, S3, and S4 Figs, respectively).

samples is not preferred for two reasons. First, classical, microscopy-based veterinary diagnostic methods such as fecal flotation and sedimentation, where eggs of parasites are identified morphologically, have relatively low sensitivity and accuracy [66]. Second, *Echinococcus* spp. eggs cannot be distinguished from those of *Taenia* spp., a closely related group of taeniid cestodes [66]. Therefore, many studies use molecular methods such as qPCR to determine presence of *Echinococcus* spp. [37,57,67]. In the last decade, however, the advent of high-throughput sequencing methods such as DNA metabarcoding, shotgun metagenomics, and restriction-site associated sequencing (RAD-seq) has resulted in a marked methodological shift in parasitology studies from wildlife to humans to domestic animals [68–72]. Indeed, DNA metabarcoding is now arguably the most popular molecular technique for biodiversity monitoring across all kinds of eco-logical studies [73,74]. High-throughput sequencing can be used to identify multiple unrelated taxa at once, which is highly advantageous in studies of helminths due to the highly specialized expertise needed for morphological identification [75]. The field of wildlife and disease ecology has therefore, in recent years, seen increased interest in assessing entire para-site communities [76–80]. These studies often use environmental DNA samples collected from the field (e.g., feces, water) due to their ease of collection compared to animal tissue samples or GI tract contents.

Though fecal samples contain PCR inhibitors that can interfere with amplification, commercial DNA isolation kits are thought to provide sufficient purification for presence/absence detection of helminths [81]. Some studies isolate eggs of helminths through fecal floats and extract DNA directly from eggs, but high-throughput sequencing techniques such as DNA metabarcoding are specialized for low-copy DNA from environmental samples, especially when paired with DNA purification methods such as freeze-thaw cycles, bead-beating, and/or commercial purification kits. Metabarcoding of fecal

or other environmental samples is increasingly used in helminth surveillance studies without the manual isolation of eggs prior to extraction protocols [65,82–84]. These methods have also been used for qPCR assays of fecal samples, as they are significantly more sensitive than fecal floats even without high-throughput sequencing [67,85]. However, we found that DNA metabarcoding of fecal samples collected from the rectum was less sensitive than either morphological confirmation of adult worms or DNA metabarcoding from a swab of small intestine contents where adult worms reside. However, we did not use additional mechanical disruption steps, such as bead-beating, which probably limited DNA recovery from our samples [57]. We would expect sequencing of field-collected scats using the same methods to be even less sensitive than samples from carcasses frozen immediately after death due to degradation from environmental factors such as rain and heat [81].

While we did not identify *E. multilocularis* in our field-collected scat samples, we did identify its close relative *Taenia pisiformis* in many of the same samples. This discrepancy could simply be due to lower prevalence or lack of presence of *E. multilocularis* in the southern part of our study area, but we did not know the true infection status of these individuals. The lack of amplification could also be due to the difference in extraction methods between our carcass fecal samples and our field-collected scats. In this case, the lack of *E. multilocularis* in field-collected samples could also be attributed to varying egg viability in outdoor conditions between *Echinococcus* spp. and *Taenia* spp. or varying resistance to freeze-thaw cycles, despite the two species having egg morphology indistinguishable to the human eye [86]. *E. multilocularis* is present as far north as the Arctic region, while *Taenia pisiformis* (the other taeniid species we detected) is a subarctic parasite, which might explain why *E. multilocularis* eggs are more freeze-resistant. Additionally, despite their eggs being morphologically identical, *Taenia* adults are much larger than *Echinococcus* adults (up to 20 m vs. only a few millimeters), so adults may shed more DNA into scat samples than *Echinococcus* spp. On the other hand, *Taenia* infection loads are typically much lower than *Echinococcus* spp. (typically in the dozens for *Taenia* and in the thousands or even tens of thousands for *Echinococcus* [52,87]). To our knowledge, no studies have directly compared DNA degradation of eggs or DNA shedding of adult worms in *Taenia* spp. and *Echinococcus* spp.; this could be an interesting area of future research.

Based on our results, we conclude that studies extracting DNA from fecal samples, even with high-throughput sequencing, may underestimate the prevalence of *E. multilocularis* and other species in the genus *Echinococcus* - especially when not using extraction methods targeting tough taeniid eggs. Our false negatives from fecal samples collected from carcasses that were promptly frozen after death suggest that even before experiencing degradation from environmental conditions, results from DNA metabarcoding of fecal samples should be interpreted with caution. It is often much more challenging to collect whole GI tracts than field-collected scat samples, especially for cryptic wildlife species. When a GI tract can be obtained, however, simply opening it and swabbing the contents may be preferable to the SFCT dissection protocol when optimizing for time and effort — especially as costs of metabarcoding continue to fall. Of course, the ensuing lab work for DNA metabarcoding is also time-consuming and expensive, but our personal observation is that GI tract dissection and morphological identification becomes far more difficult as sample size and number of species of interest increase. One major disadvantage of metabarcoding is that it cannot accurately quantify infection load within an individual. However, populations of *Echinococcus* spp. in infected individuals are often in the tens of thousands [52]. While methods exist to subsample and extrapolate infection load [52], we could not find any studies thoroughly validating these counting methods, likely due to the time it would take to count entire populations of *Echinococcus* repeatedly. Large-scale monitoring studies using GI tracts should therefore consider collecting and sequencing intestinal swabs to monitor *E. multilocularis* rather than conducting SFCT dissections, especially if researchers are interested in detecting multiple parasite species and if quantification of parasite load is not a priority. Indeed, the primary strength of DNA metabarcoding is the ability to identify coinfections of multiple helminth species, which are common in wildlife and reduce the efficacy of methods such as Sanger sequencing. When possible, monitoring efforts using fecal samples should consider incorporating multiple diagnostic techniques to avoid false negatives that could obscure the true distribution of this parasite [67].

## Transmission dynamics and host ecology of *E. multilocularis*

Based on our study and others in recent months confirming the presence of *E. multilocularis* in western North America, *E. multilocularis*' range may be much larger than previously known. Across North America, infection prevalence of *E. multilocularis* in red fox, wolf, and coyote populations is thought to be relatively low [37,85,88–90]. Our results suggest a much higher prevalence rate than most other similar studies, with the exception of coyote populations in highly endemic regions of the provinces of Saskatchewan and Alberta, Canada [67,89,90,64]. Notably, in Edmonton, Alberta, coyotes from urban areas had significantly higher prevalence than coyotes from rural areas [89].

While AE is still rare in temperate regions, shifting wildlife populations and increasing urbanization may have facilitated the expansion of *E. multilocularis* into previously unaffected areas [1,9,24,87,91,92]. Its primarily sylvatic life cycle using highly mobile hosts may have allowed *E. multilocularis* to expand rapidly and remain unnoticed due to the intensive labor required to conclusively identify it (Santa et al. 2023). In Europe, red foxes, raccoon dogs, and recently jackals are the most common hosts of *E. multilocularis* [16,17]. Though recent studies on red and gray foxes are limited, coyotes seem to be the most common host for *E. multilocularis* in North America [12,26,57], with gray wolves only rarely implicated [93]. Still, gray wolves may be ecologically important hosts - due to their large home range sizes and dispersal distances compared to other North American canids, they may serve to connect disparate populations of *E. multilocularis* [19].

There are several potential introduction mechanisms that may have contributed to the difficulty of tracking the spread of *E. multilocularis*. The genetic discontinuity from sequenced worms from "European-type" worms throughout North America support the theory of multiple founder events (Santa et al. 2023). One common theory is that, starting in the mid-1700s, red foxes introduced to the western US from Europe for hunting use were infected with *E. multilocularis* and able to quickly spread it in a naive environment [1]. Similarly, captive red foxes that escaped from commercial fur farms, popular at the turn of the 20th century, may have played a similar role [94,95]. The reintroduction of wolves into the western U.S. over the last several decades may have also facilitated the introduction of *E. multilocularis* populations from relocated wolves' native ranges – though transplanted wolves were treated with praziquantel, parasitologists have speculated that some individual cestodes may have survived treatment [96]. Lastly, domestic dogs imported from Europe or Asia may carry worms into North America - perhaps surprisingly, both Canada and the USA do not require deworming treatment of dogs prior to import [1].

According to DNA metabarcoding data from a concurrent study [58], coyotes in western Washington consume a variety of rodent species, including long-tailed voles (*Microtus longicaudus*), Norway rats (*Rattus norvegicus*), black rats (*Rattus rattus*), gray squirrels (*Sciurus caroliniensis*), and North American beavers (*Castor canadensis*). Cricetid rodents are the most common intermediate hosts in North America, with deer mice (*Peromyscus* spp.) and voles (*Microtus* spp.) thought to be responsible for the vast majority of transmission [19], but other rodents such as muskrats (*Odantra zibethicus*) and chipmunks (family Sciuridae) have also been implicated [25]. The extent of viable intermediate host species in North America is still unclear and constantly changing [8,97]. Due to this lack of information, and since the coyotes in our study area consume a large variety of rodents, it is difficult to know which species serve as intermediate hosts in our study region. Further, *E. multilocularis* is thought to have highly spatially clustered infections in its intermediate hosts [1], which may explain the clustering of our positive coyote samples (Fig 3). Meadow voles (*Microtus pennsylvanicus*) and southern red-backed voles (*Clethrionomys gapperi*) are both common intermediate hosts for *E. multilocularis* [25,98], while long-tailed voles - which are more common in western Washington and the only voles found in our diet metabarcoding data - have not yet been confirmed as viable hosts. Similarly, Norway rats have been confirmed as intermediate hosts in both Europe [99] and Asia [100], but we could not find any records of *E. multilocularis* in rats in North America - perhaps because they are more likely to be found in dense urban environments, and few studies exist targeting surveillance of intermediate hosts of helminths in these areas due to their inability to infect humans at this stage. Since the life cycle of *E. multilocularis* does not

involve livestock species like its close relative *E. granulosus* and infections may not be apparent in wild rodents even in live capture studies, presence of *E. multilocularis* in intermediate hosts may go unnoticed except in studies focusing specifically on helminth infections in wild rodents. Further studies on prevalence in intermediate hosts, including competency of new potential intermediate host species, should be conducted [97]. While coyotes are generalists and likely eat more rodent species than were captured by our diet data, the quickly expanding literature on both intermediate and definitive hosts highlights the necessity of surveilling closely related species of known hosts [8,25,97,101].

### Public health implications and recommendations

To the best of our knowledge, there have been no reports of AE in humans on the west coast of North America (outside of Alaska), but recent diagnoses of intestinal and AE in domestic dogs raise concern [33,40]. Dog ownership is a risk factor for human AE as dogs interact closely with humans and shed feces into the environment [102]. Dog owners and their veterinarians can reduce transmission risk by not allowing their dogs to consume soil or feces, and by practicing regular deworming treatments with praziquantel-containing products, especially if their dog exhibits coprophagy, geophagia, or pica, or if the dog kills and consumes wild rodents [29]. Additionally, the potential for introduction through pet travel and international dog adoption raises concerns for emerging transmission hotspots in new geographic regions [103–105,64].

The early stages of AE in humans often show no symptoms, with the disease remaining symptomless for 10–15 years or longer [21]. Therefore, most patients do not present until symptoms are advanced and hydatid cysts are extremely difficult to extract. In our study region, vulnerable populations include immunocompromised people, who might serve as sentinels for the disease due to shorter incubation periods [11]. Unhoused individuals may also be at higher risk due to increased barriers to accessing healthcare, especially if they share outdoor spaces with coyotes and particularly if they own dogs [106–108]. Treatment for late-stage disease involves surgery [109], and pharmacological treatment is still extremely limited [110]. Further, knowledge of AE is limited amongst medical professionals outside of countries where it is highly endemic. However, the known range of *E. multilocularis* is consistently expanding to areas that are previously thought to be free of *E. multilocularis* populations, and our paper adds to the building evidence that *E. multilocularis* is increasingly overlapping with dense human populations [64]. It is therefore crucial for public health and medical professionals across the Northern Hemisphere to be aware of the presence of *E. multilocularis* in their regions in order to expedite accurate diagnoses. Indeed, the recent increase of human AE cases in North America may be due in part to increased awareness of the disease among health professionals [64]. Educational programs targeted towards those with higher risk for AE should be implemented in areas where *E. multilocularis* has been discovered, such as gardeners, trappers, unhoused individuals, and dog owners. These programs could encourage individuals to wear gloves, practice hand hygiene, thoroughly wash foraged fruit, vegetables, or fungi, and/or regularly deworm their pets with praziquantel or similar drugs.

Currently, echinococcosis is not federally reportable in either the U.S. or Canada, but it recently became a notifiable disease for humans in the state of Washington as of December 2024 [111]. It is also a notifiable disease for dogs and other animals in the state of Washington [112]. Despite this legislation, monitoring and public education are still extremely limited, and reports of *E. multilocularis* across western North America are isolated. It is impossible to know whether or not this scarcity is due to populations of the parasite being rare and clustered or due to a lack of surveillance, though several studies have posited that the latter is indeed probable [1,113]. Since AE is rare and difficult to diagnose, it is crucial for public health officials, medical professionals, and veterinary professionals to be aware of its existence in a differential diagnosis. The United States and Canada should therefore consider implementing transboundary surveillance programs, as has been done in China and across the European Union [114], and should also make echinococcosis a federally reportable disease for both humans and pets.

## Supporting information

**S1 Table. Positive control samples and GenBank sequences used to test our custom cestode primer.**
(DOCX)

**S2 Table. Results of analysis comparing sequencing success of Illumina DNA sequencing for Taenia pisiformis and Echinococcus multilocularis between three sample types: intestinal swabs, fecal matter (200mg), and fecal swabs.** The last row indicates the proportion of the number of samples in which the species was detected in that method (i.e., fecal swab or fecal matter) over the number of samples in which the species was detected using intestinal swabs, using our sequence read cutoff of 1000. For example, T. pisiformis was detected in 13 intestinal swabs, but only detected in 5 fecal swabs (with a read cutoff of 1000); therefore, detection success for this method was 0.38.
(DOCX)

**S1 Data. Fasta file showing alignments of concatenated sequences of cestodes used to develop our primer.**
Sequences include those extracted from positive control samples and publicly available sequences on GenBank (see S1 Table).
(FAS)

**S2 Data. Fasta file showing alignments of concatenated *E. multilocularis* sequences (cob, nad2, and cox1); sequences include samples from our study as well as several haplotype sequences from various global *E. multilocularis* studies (see Fig 4).**
(FAS)

**S1 Fig. A map of Europe and Asia depicting the spread of E. multilocularis across the last several decades.**
Each color represents a different decade wherein a peer-reviewed study first confirmed the presence of E. multilocularis morphologically, molecularly, or both. In some endemic regions of Russia, Mongolia, and China, where it was difficult to locate peer-reviewed studies in English, we referenced maps from previously published meta-analyses on E. multilocularis range [115,19]. In cases where only a few isolated reports exist (e.g., northern Italy), polygons have been expanded slightly beyond the actual range of detection for visibility. The base map is available via Adobe Stock at the following link: https://stock.adobe.com/images/detailed-world-map-with-borders-of-states-isolated-world-map-isolated-on-white-background-vector-illustration/487393684.
(PNG)

**S2 Fig. Unmanipulated (stitched) photo of unstained *E. multilocularis* slide (shown in Fig 5A) from Leica imaging software.**
(JPG)

**S3 Fig. Unmanipulated photo of stained *E. multilocularis* slide (shown in Fig 5B) from Leica imaging software.**
(PNG)

**S4 Fig. Unmanipulated photo of stained *E. multilocularis* slide (shown in Fig 5C) from Leica imaging software.**
(PNG)

**S1 Text. Methods for data sourcing and creation of Figs 1 and S1.**
(DOCX)

**S2 Text. Links to Youtube videos of modified SFCT gastrointestinal dissection protocol developed for this manuscript, which include swabbing for genetics samples and a shortened double-sieving process.** A text description of the protocol is available in the Methods section.
(DOCX)

## Acknowledgments

We are grateful to the following collaborators for their carcass donations and assistance: E. Meredith and colleagues at the Progressive Animal Welfare Society in Monroe, Washington; M. Stevens, D. Taylor, and colleagues at the United States Department of Agriculture - Wildlife Services division; and M. Blankenship, K. West, and colleagues at the Washington Department of Fish and Wildlife. We are also grateful to J. Bradley at the Burke Museum of Natural History and Culture and K. Leslie of the Wood lab for their assistance in facilitating laboratory work and guidance in laboratory techniques. We would like to thank E. Baker and the Gerhold lab at the University of Tennessee - Knoxville for providing positive control samples for our study; K. Buck Garrett and the Cleveland lab at the University of Georgia for assistance in developing an intestinal dissection protocol and for providing positive control samples; and K. Haman and K. Mansfield of the Washington Department of Fish and Wildlife for their help in determining previous *Echinococcus* spp. detections in Washington. Finally, we thank the dozens of undergraduate students and volunteers who dedicated their time to assisting in data collection and lab work for this study.

## Author contributions

**Conceptualization:** Yasmine Hentati, Laura R. Prugh, Christopher J. Schell, Chelsea L. Wood.

**Data curation:** Yasmine Hentati, Ellie Reese, Claire C. Curran, Erika M. Miller, Dakeishla M. Díaz-Morales, Samantha E.S. Kreling.

**Formal analysis:** Yasmine Hentati, Ellie Reese, Claire C. Curran, Erika M. Miller.

**Funding acquisition:** Yasmine Hentati, Laura R. Prugh, Christopher J. Schell, Chelsea L. Wood.

**Investigation:** Yasmine Hentati, Ellie Reese, Claire C. Curran, Erika M. Miller, Dakeishla M. Díaz-Morales, Samantha E.S. Kreling.

**Methodology:** Yasmine Hentati, Ellie Reese, Claire C. Curran, Erika M. Miller, Dakeishla M. Díaz-Morales, Samantha E.S. Kreling, Guilherme G. Verocai, Chelsea L. Wood.

**Project administration:** Yasmine Hentati, Laura R. Prugh, Christopher J. Schell, Chelsea L. Wood.

**Resources:** Ellie Reese, Guilherme G. Verocai, Chelsea L. Wood.

**Supervision:** Laura R. Prugh, Christopher J. Schell, Chelsea L. Wood.

**Validation:** Yasmine Hentati, Ellie Reese, Claire C. Curran, Erika M. Miller, Dakeishla M. Díaz-Morales, Samantha E.S. Kreling, Guilherme G. Verocai.

**Visualization:** Yasmine Hentati, Samantha E.S. Kreling.

**Writing – original draft:** Yasmine Hentati.

**Writing – review & editing:** Yasmine Hentati, Ellie Reese, Claire C. Curran, Erika M. Miller, Dakeishla M. Díaz-Morales, Samantha E.S. Kreling, Guilherme G. Verocai, Laura R. Prugh, Christopher J. Schell, Chelsea L. Wood.

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
