## [Decision Letter · Decision Letter 0]

3 Nov 2025

*Echinococcus multilocularis*
Response to Reviewers
Revised Manuscript with Track Changes
Manuscript

Shaden Kamhawi

co-Editor-in-Chief

Paul Brindley

co-Editor-in-Chief

**Additional Editor Comments:**
**Journal Requirements:**

4) We are unable to open the following Supporting Information file: supporting_information.zip. Please kindly revise as necessary and re-upload.

Potential Copyright Issues:

- Figure 3. Please confirm whether you drew the images / clip-art within the figure panels by hand. If you did not draw the images, please provide (a) a link to the source of the images or icons and their license / terms of use; or (b) written permission from the copyright holder to publish the images or icons under our CC BY 4.0 license. Alternatively, you may replace the images with open source alternatives. See these open source resources you may use to replace images / clip-art:

- Figures 1, 2, and 4. Please (a) provide a direct link to the base layer of the map (i.e., the country or region border shape) and ensure this is also included in the figure legend; and (b) provide a link to the terms of use / license information for the base layer image or shapefile. We cannot publish proprietary or copyrighted maps (e.g. Google Maps, Mapquest) and the terms of use for your map base layer must be compatible with our CC BY 4.0 license.

6) Please ensure that the funders and grant numbers match between the Financial Disclosure field and the Funding Information tab in your submission form. Note that the funders must be provided in the same order in both places as well.

State what role the funders took in the study. If the funders had no role in your study, please state: "The funders had no role in study design, data collection and analysis, decision to publish, or preparation of the manuscript.".

**Reviewers' comments:**

**Key Review Criteria Required for Acceptance?**

**Methods:**

-Are the objectives of the study clearly articulated with a clear testable hypothesis stated?

-Is the study design appropriate to address the stated objectives?

-Is the population clearly described and appropriate for the hypothesis being tested?

-Is the sample size sufficient to ensure adequate power to address the hypothesis being tested?

-Were correct statistical analysis used to support conclusions?

-Are there concerns about ethical or regulatory requirements being met?

Reviewer #1: - Section starting on 310: Why was only one worm from each positive carcass used for haplotype identification? Did you check the amplified DNA markers for overlapping peaks that could represent mixed infections? Homogenizing pooled samples of a subset of worms from a carcass and sequencing that DNA could reveal if there were multiple haplotypes present in an infected individual.

- Line 388 – what do you mean by 6 coyotes- do you mean 6 worms isolated from 6 different infected coyotes? Why did you use only 6 if you had 37 positive individuals? Why do you think 2 didn’t work- were they not infected with E. multilocularis? If the DNA was extracted from worms isolated from infected coyotes, and the PCR didn’t work on DNA from these worms- are these worms not E. multilocularis? Were there any other steps taken to identify these DNA/these worms?

- Line 388-392: What gene was sequenced? Please include accession numbers for the haplotypes listed in figure 5 and the accession number for the gene you generated. Did you generate sequences from all 3 target genes (nad2, cob, cox1) or just one?

- Line 392: Figure 5, not 3.

Reviewer #2: See attached. The DNA extraction method used for fecal analysis was not optimal (would not have cracked the eggs), leading to false negatives, and concluding that invasive intestinal sampling is the only way to sensitively detect DNA of E. multilocularis in wild canids is not supported (and not a justified animal use). The appropriate conclusion is that DNA extraction methods optimized for dietary metabarcoding and host genetic analysis are not sufficient for extracting DNA from environmentally resistant cestode eggs. Coprosurveillance for E. multilocularis in Europe and Canada offers several examples of highly sensitive DNA extraction and "DNA fishing" methods that make noninvasively collected fecals a very useful surveillance method indeed.

Reviewer #3: The study is well designed and the methods/ sample sizes are adequate for the detection of E.m. in coyotes in the study area. The carcasses in particular assist in estimating the prevalence of E.m. in these hosts.

Here are some suggestions for minor amendments to the 'Methods' chapter:

Lines 182-200: The segments 'Potential definitive hosts' and 'Potential intermediate hosts' do not belong in the 'Methods' chapter. As results of a previous study, information on the diet of Washington coyotes may be included as part of the introduction (possible wildlife hosts in Washington) or in the discussion ('Transmission dynamics…').

Line 193: Period after “(50)” should be a comma.

Lines 201-207: In which years were the carcasses collected? Please add this information.

Line 235: The G in “Gloves” is missing.

Lines 272-274: The “freshness categories” should be mentioned in the segment 'Coyote scat collection' rather than here.

Lines 310-315: Why was the haplotype analysis limited to only six positive coyotes? Is there a reason why it was it necessary to confirm E. multilocularis both morphologically and with Illumina sequencing before Sanger sequencing? Worms were present in 33 individuals – why not genetically identify them with Sanger sequencing?

**Results:**

-Does the analysis presented match the analysis plan?

-Are the results clearly and completely presented?

-Are the figures (Tables, Images) of sufficient quality for clarity?

Reviewer #1: -Figure 1, 2, and 4 look very good, are of great quality and are accurate to the literature (F1 and F2, F4 is new data). Figure 3 could be moved to the beginning of the paper because it fits the introduction better. Figure 1 is interesting, but as the paper is not focused on defining the global distribution of E. multilocularis, it could be removed.

- The data in table 2 should be split to represent:

1) Only the prevalence of E. multilocularis in samples tested (carcass and fecal samples, and how many unique individuals were represented in both sample types), giving the 37%, or 37/100 prevalence information clearly.

2) The difference between E. multilocularis identification between the morphological ID, fecal samples from carcass, and intestinal swabs from carcass. This information is confusing in the full table, and it will allow you to break up the table description. This will also provide clearer evidence for the use of different methods and best practices for E. multilocularis detection.

- Figure 6- Well done staining and picture.

Reviewer #2: Please see attached.

Reviewer #3: The results and figures are mostly well presented, though there is some lack of clarity that may be easily amended:

The field-collected scat samples were not screened for the presence of taeniid eggs via flotation, correct? It would have been interesting to note if T. pisiformis was detected in all taeniid-positive samples or if other taeniid species/ low egg counts were missed by the NGS approach.

Figures 1 & 2 (Introduction): The light-yellow colouration of the endemic areas pre 1990s is barely distinguishable from the white background. Please choose a darker tint for better visibility.

Table 2: Separating the scat results into their “freshness categories” in Table 2 seems rather unnecessary, as E. multilocularis was not detected in any of them and the freshness aspect is not further discussed (for example regarding successful Taenia detection). It might also make more sense to change the table setup for clarity, as suggested in the reviewer attachment.

Line 392: “Figure 3” should be “Figure 5”.

Figure 5: Please add the sequence lengths of the individual gene regions (cob, cox1, nad2), as well as the total length of the concatenated sequences used for the construction of the haplotype network.

**Conclusions:**

-Are the conclusions supported by the data presented?

-Are the limitations of analysis clearly described?

-Do the authors discuss how these data can be helpful to advance our understanding of the topic under study?

-Is public health relevance addressed?

Reviewer #1: (No Response)

Reviewer #2: Please see attached, and above comments about fecal testing.

Reviewer #3: Most conclusions are supported by the data presented, however, there are some uncertainties and open questions that should be addressed:

It might be interesting to deliberate (either in the introduction or in the discussion) if the increase of human AE cases in North America can also be linked to an increased awareness of this disease among health professionals (fewer misdiagnoses).

Lines 436-438: “…since at least 2021” - Why is this particular year mentioned? Is there reason to believe this transmission cycle was established this recently?

Lines 485-489: As a note from personal experience, it is rather unlikely that Taenia eggs are more resistant to freeze-thaw cycles than those of Echinococcus.

Line 550: “(Massolo et al., 2014)” should be “(1)”.

Lines 564-566: This is perhaps a misunderstanding. Muskrats are not thought to be the ecologically most important intermediate hosts in Europe, where they are an invasive species. They exhibit the highest prevalence for E. m.; however, muskrats are restricted to only a few habitats in Europe, and they are not as common and widespread (and a part of the diet of foxes) as, for example, voles.

**Editorial and Data Presentation Modifications?**

Reviewer #1: - Lines 69-72. These two sentences are not linked clearly (spread in Europe � canid distribution). Also, lines 70 and 72, “Given the geographically expansive distributions of canid species – the definitive host of E. multilocularis – zoonotic spillover commonly occurs to humans and their pets (19,20).” – this is not accurate, E. multilocularis spillover to humans and pets is quite rare, and has much to do with the prevalence of E. multilocularis in canids versus just the available host distribution. Be clear with the data from the two papers (Davidson et al. 2016 and Crotti et al. 2023) that is being used to support this.

- Paragraph that spans lines 82-95: split up to discuss range in one paragraph and hosts in another.

- Citation needed line 100

- Line 182, Potential definitive hosts: missing Latin name for wolf species in the area

- Line 235: Gloves

- In the discussion- the citation style changes from numbers to full citations, please keep consistent.

Reviewer #2: Please see attached.

Reviewer #3: (No Response)

**Summary and General Comments**

Reviewer #1: This is a very important and relevant study on the expansion and characterization of Echinococcus multilocularis in a novel geographic range in North America. The authors conducted a robust surveillance study and discovered a high prevalence of E. multilocularis in wildlife in a region lacking previous surveillance efforts. This is highly significant as this parasite can cause a fatal disease in humans, and is currently underreported and misdiagnosed, even in endemic areas. Continued surveillance, reporting, and education is vital to understanding and controlling this parasite, especially as emerges in novel areas within the United States.

This study is well-written, with very well-done figures, and detailed conclusions on the work. A few structural changes should be made to the introduction to keep it focused and organized to E. multilocularis knowledge in North America, where the study takes place. More information is needed on the molecular methodology and results, and accession numbers for the genes used to identify the haplotypes detected in this study should be included, so that they can be openly compared to the haplotypes being detected across North America. Specific comments and questions addressed in the sections above. I recommend that this publication be accepted with minor revisions.

Reviewer #2: Please see attached. The introduction and discussion review the literature rather than speak to this study, and should be significantly reduced and focused on the rationale and significance of this study. No new experiments are needed, just more careful interpretation of the findings.

Reviewer #3: This nicely written manuscript provides new data on the presence and spread of E. multilocularis in coyotes in Washington State (USA) and highlights the implications for alveolar echinococcosis risk in the Pacific Northwest.

Gastrointestinal tracts obtained from coyote carcasses, as well as field-collected scats were examined for Echinococcus worms and eggs. DNA was extracted from faecal samples, intestinal swabs, and tapeworm material for molecular analysis and E. multilocularis haplotype identification. Individual coyotes were identified using microsatellite markers. As a byproduct, the sensitivity of morphological and molecular detection methods was evaluated and compared.

In total, almost half (45.7%) of the 81 examined coyote carcasses were infected with E. multilocularis. The E. multilocularis haplotype identified in four individual coyotes was previously reported from British Columbia (Canada) and is closely related to haplotypes from France. None of the 30 field-collected scat samples tested positive for E. multilocularis, which might be due to a methodical error. This is briefly discussed.

This manuscript is a proper piece of work which confirms that E. multilocularis has an established sylvatic life cycle in Washington State.

PLOS authors have the option to publish the peer review history of their article (what does this mean? ). If published, this will include your full peer review and any attached files.

**Do you want your identity to be public for this peer review?** For information about this choice, including consent withdrawal, please see our Privacy Policy .

Reviewer #1: No

Reviewer #2: No

Reviewer #3: No

**Figure resubmission:**

**Reproducibility:** To enhance the reproducibility of your results, we recommend that authors of applicable studies deposit laboratory protocols in protocols.io, where a protocol can be assigned its own identifier (DOI) such that it can be cited independently in the future. Additionally, PLOS ONE offers an option to publish peer-reviewed clinical study protocols. Read more information on sharing protocols at https://plos.org/protocols?utm_medium=editorial-email&utm_source=authorletters&utm_campaign=protocols

---

## [Decision Letter · Decision Letter 1]

6 Feb 2026

Dear Dr. Hentati,

We are pleased to inform you that your manuscript 'Detection of *Echinococcus multilocularis* in coyotes in Washington State, USA highlights need for increased wildlife surveillance' has been provisionally accepted for publication in PLOS Neglected Tropical Diseases.

Best regards,

Majid Fasihi Harandi, PhD

Academic Editor

Uriel Koziol

Section Editor

Shaden Kamhawi

co-Editor-in-Chief

Paul Brindley

co-Editor-in-Chief

Reviewer's Responses to Questions

**Key Review Criteria Required for Acceptance?**

**Methods**

-Are the objectives of the study clearly articulated with a clear testable hypothesis stated?

-Is the study design appropriate to address the stated objectives?

-Is the population clearly described and appropriate for the hypothesis being tested?

-Is the sample size sufficient to ensure adequate power to address the hypothesis being tested?

-Were correct statistical analysis used to support conclusions?

-Are there concerns about ethical or regulatory requirements being met?

Reviewer #1: The authors addressed all concerns regarding methodology, which were mainly focused on clarifying some of the decisions made during screening and sequencing.

Reviewer #3: The methods are explained in detail and are appropriate for the objectives of this study.

**Results**

-Does the analysis presented match the analysis plan?

-Are the results clearly and completely presented?

-Are the figures (Tables, Images) of sufficient quality for clarity?

Reviewer #1: The results shown in Table 2 and Table 3 are much clearer.

Reviewer #3: The tables and figures have greatly improved since the last version, they are clear and comprehensible.

**Conclusions**

-Are the conclusions supported by the data presented?

-Are the limitations of analysis clearly described?

-Do the authors discuss how these data can be helpful to advance our understanding of the topic under study?

-Is public health relevance addressed?

Reviewer #1: (No Response)

Reviewer #3: The conclusions are supported by the results of this study. The authors emphasize the public health relevance of the findings.

**Editorial and Data Presentation Modifications?**

Reviewer #1: (No Response)

Reviewer #3: This manuscript has been greatly enhanced since its last revision. In its current form it is suitable for publication.

**Summary and General Comments**

Reviewer #1: Really excellent work! The authors addressed all editors' comments and have strengthened the manuscript by streamlining the main findings and information and clarifying their methodology and protocols. This study provides another piece of evidence for the expansion of E. multilocularis in North America and the need for increased surveillance in wildlife. In conjunction with their novel findings, the authors also provide a robust review of the state of E. multilocularis in North America, as well as the public health implications of this parasite's range expansion and the current limitations and difficulties for conducting surveillance for E. multilocularis. All this together will provide readers with a strong understanding of the current state of E. multilocularis in North America and the relevance of their discoveries in Washington.

Reviewer #3: (No Response)

PLOS authors have the option to publish the peer review history of their article (what does this mean? ). If published, this will include your full peer review and any attached files.

**Do you want your identity to be public for this peer review?** For information about this choice, including consent withdrawal, please see our Privacy Policy .

Reviewer #1: No

Reviewer #3: No

---

## [Editor Report · Acceptance letter]

Dear Dr. Hentati,

We are delighted to inform you that your manuscript, "Detection of *Echinococcus multilocularis* in coyotes in Washington State, USA highlights need for increased wildlife surveillance," has been formally accepted for publication in PLOS Neglected Tropical Diseases.

Best regards,

Shaden Kamhawi

co-Editor-in-Chief

Paul Brindley

co-Editor-in-Chief
